# A MINIMALIST DATASET FOR SYSTEMATIC GENERALIZATION OF PERCEPTION, SYNTAX, AND SEMANTICS

**Qing Li**[1], **Siyuan Huang**[1], **Yining Hong**[2], **Yixin Zhu**[3], **Ying Nian Wu**[2], **Song-Chun Zhu**[1,2,3]
[1]National Key Laboratory of General Artificial Intelligence, BIGAI
[2]Center for Vision, Cognition, Learning, and Autonomy (VCLA), UCLA
[3]Institute for Artificial Intelligence, Peking University
`https://liqing-ustc.github.io/HINT`

## ABSTRACT

Inspired by humans' exceptional ability to master arithmetic and generalize to new problems, we present a new dataset, Handwritten arithmetic with INTegers (HINT), to examine machines' capability of learning generalizable concepts at three levels: *perception*, *syntax*, and *semantics*. In HINT, machines are tasked with learning how concepts are perceived from raw signals such as images (*i.e.*, perception), how multiple concepts are structurally combined to form a valid expression (*i.e.*, syntax), and how concepts are realized to afford various reasoning tasks (*i.e.*, semantics), all in a weakly supervised manner. Focusing on systematic generalization, we carefully design a five-fold test set to evaluate both the *interpolation* and the *extrapolation* of learned concepts w.r.t. the three levels. Further, we design a few-shot learning split to determine whether or not models can rapidly learn new concepts and generalize them to more complex scenarios. To comprehend existing models' limitations, we undertake extensive experiments with various sequence-to-sequence models, including RNNs, Transformers, and GPT-3 (with the chain of thought prompting). The results indicate that current models struggle to extrapolate to long-range syntactic dependency and semantics. Models exhibit a considerable gap toward human-level generalization when evaluated with new concepts in a few-shot setting. Moreover, we discover that it is infeasible to solve HINT by merely scaling up the dataset and the model size; this strategy contributes little to the extrapolation of syntax and semantics. Finally, in zero-shot GPT-3 experiments, the chain of thought prompting exhibits impressive results and significantly boosts the test accuracy. We believe the HINT dataset and the experimental findings are of great interest to the learning community on systematic generalization.

## 1 INTRODUCTION

Humans possess a versatile mechanism for learning concepts from data (Firestone & Scholl, 2016). Suppose, for example, that we were tasked with deciphering ancient Egyptian signs based on the examples in Table 1. Given sufficient time, we may comprehend these signs by how to recognize them—what each sign looks like at the *perceptual* level, by how to compose them into valid sequence—at the *syntactic* level, and how to predict the results—at the *semantic* level. Learning concepts heavily rely on these three-level interweaving meanings. Such observation is also consistent with the classic view of human cognition, which postulates at least three distinct levels of organizations in computation systems (Pylyshyn, 1984).

| Train | | Test |
|---|---|---|
| ◦𝕝ʚ∪𝕝●● → 60 | ○∪⅂○ʚ◿◿ → 18 | ◦𝔸∪🕮ʚ𝕤𝕤🕮𝕝◿◿ → ? |
| ⅂◦🕮𝕤◿∪● → 100 | ∪◦◿🕮⅂ʚʚ → 16 | 𝕝𝕝⅂◦🕮𝕤◿◿ → ? |
| ∪⅂𝕤∪◿ → 12 | ∪𝕝🕮●🕮◿🕮●◿ → 41 | 🕮𝔸◦𝕝●◿🕮𝕤ʚ → ? |
| 𝔸🕮◦◿ʚ🕮◿ → 4 | 🕮∪𝕤𝔸●𝔸◿ → 4 | 𝕝🕮ʚ◦🕮𝕤○∪🕮◿◿●𝕝∪◦🕮ʚ◿◿ → ? |
| 🕮🕮ʚ◦🕮●● → 26 | ○∪◦𝕤● → 17 | 𝔸🕮●◦🕮●🕮🕮◿◦𝔸◦●●🕮🕮ʚ◿◿🕮◿ → ? |

Table 1: **Can you decipher these ancient Egyptian signs from training examples and apply them to test cases?** Interested readers can refer to the website, `https://liqing-ustc.github.io/HINT/Egyptian`, for more training and test samples with the ground-truth meaning for each sign. We strongly encourage the readers to play this game prior to reviewing the answers.

Another appealing characteristic of human concept learning is its *systematic compositionality* (Chomsky, 1957; Montague, 1970): the algebraic capacity to understand and construct an endless number of novel combinations from a finite set of known components, *i.e.*, "infinite use of finite means" (Chomsky, 1965). As illustrated in Table 1, this form of compositionality is essential to the human ability to make strong generalizations from simple examples to complex ones.

Various benchmarks (Lake & Baroni, 2018; Hupkes et al., 2020; Keysers et al., 2020) and methods (Lake, 2019; Gordon et al., 2019; Csordás et al., 2021) have been introduced by the emerging community of learning models that capture human-like systematic compositionality. As it is difficult to collect real data with systematic compositionality, the majority of existing benchmarks are derived from artificial domains using synthetic data and tasks, covering only a subset of the concept learning spectrum; see Table 2 for a detailed comparison. When evaluating systematic compositionality, prior datasets frequently conflate syntax and semantics. For instance, the SCAN dataset (Lake & Baroni, 2018) is a semantic parsing task from natural language commands to action sequences; when a model fails on a longer command than the ones in the training set, the root cause could stem from misinterpreting the complex syntactic relations in a long input sequence (command) or its inability to generate a long output sequence (actions) (*e.g.*, as a result of the EOS decision problem (Newman et al., 2020). In addition, previous benchmarks frequently incorporated simple semantics (*e.g.*, a simple mapping or repetition), resulting in an undesired bias toward syntactic generalization.

To expand systematic compositionality to a full-spectrum systematic generalization w.r.t. perception, syntax, and semantics, we draw inspiration from arithmetic and present a new benchmark called HINT, Handwritten arithmetic with INTegers. The HINT task is intuitive: Machines accept as input *images* of handwritten expressions and predict the final results of expressions, restricted in the integers. Since there is no intermediary supervision, the three-level meanings are apparently intertwined during learning, and models are expected to simultaneously acquire the three-level meanings to make correct predictions. To provide a comprehensive and rigorous test of how models generalize the learned concepts, we introduce a carefully structured evaluation scheme with five subsets, focusing on generalization patterns (*i.e.*, interpolation and extrapolation) at various levels (*i.e.*, perception, syntax, and semantics). In addition, we build a few-shot learning split to determine if models can rapidly learn new concepts from few examples and generalize them to more complicated scenarios. Being minimal yet comprehensive in terms of systematic generalization, HINT is fundamentally more difficult than earlier datasets because: (i) The images are of actual handwriting with considerable visual variation; (ii) The syntactic relations between the tokens in the expressions are more complex with long-range dependency. (iii) The semantics of arithmetic concepts are more complex than the simple mappings in prior datasets.

To facilitate future research in this direction, we conduct extensive experiments of various sequence-to-sequence (seq2seq) models, including Recurrent Neural Networks (Hochreiter & Schmidhuber, 1997; Chung et al., 2014), Transformers (Vaswani et al., 2017), and GPT-3 (Brown et al., 2020) (with chain of thought prompting Wei et al. (2022)). Our experiments indicate that all models still struggle on HINT; even the state-of-the-art model, Universal Transformer (Dehghani et al., 2018) with relative positional encoding (Shaw et al., 2018; Dai et al., 2019), achieves just 54% accuracy on HINT, although it achieves virtually perfect accuracy on prior datasets such as SCAN (Csordás et al., 2021). An in-depth analysis of the results on each test subset reveals that current models still struggle with extrapolation to long-range syntactic dependency and semantics. In the GPT-3 experiments, the chain of thought prompting significantly increases the zero-shot test accuracy from 8.6% to 27.6%. By examining the scaling trends of the test accuracy w.r.t. the size of the model and the dataset, we find that it is impractical to solve HINT by simply scaling up the size of the dataset or the model, as is typically done in NLP tasks (Kaplan et al., 2020; Henighan et al., 2020); more data and parameters do not significantly improve the extrapolation over syntax and semantics. The few-shot learning experiments demonstrate that, despite the fact that the top-performing models exhibit decent capabilities for learning new concepts, they are still far from the human-level generalization that only requires the learning examples of a new concept in a primitive form and readily generalizes to more complex compositions of the learned concept.

In short, we introduce the HINT dataset for investigating the systematic generalization across three levels—perception, syntax, and semantics. By benchmarking various seq2seq models on HINT, we uncover their primary weaknesses in systematic generalization. We hope the HINT dataset and our experimental findings will stimulate future developments of systematic generalization.

Table 2: **Dataset categorization and comparison.** SP: semantic parsing, IC: image classification, QA: question answering, i&t: image & text. Perception/Syntax/Semantics: whether the task requires models to learn perception/syntax/semantics. Generalization: the type of generalization required for test examples. *: the generated images in these datasets have little variance.

| Dataset | Domain | Task | Modality | Perception | Syntax | Semantics | Generalization | Size |
|---|---|---|---|---|---|---|---|---|
| SCAN (Lake & Baroni, 2018) | synthetic | SP | text | | ✓ | ✓ | systematic | 100K |
| gSCAN (Ruis et al., 2020) | synthetic | SP | i&t | ✓* | ✓ | ✓ | systematic | 300K |
| PCFG (Hupkes et al., 2020) | synthetic | SP | text | | ✓ | ✓ | systematic | 100K |
| CFQ (Keysers et al., 2020) | real | SP | text | | ✓ | ✓ | systematic | 239K |
| CURI (Vedantam et al., 2021) | synthetic | IC | image | ✓ | | ✓ | systematic | 15K |
| COGS (Kim & Linzen, 2020) | real | SP | text | | ✓ | ✓ | systematic | 30K |
| Mathematics (Saxton et al., 2018) | real | QA | text | | ✓ | ✓ | systematic | 2M |
| PGM (Barrett et al., 2018) | synthetic | IC | image | ✓ | | ✓ | systematic | 1.4M |
| CLOSURE (Bahdanau et al., 2019) | synthetic | QA | i&t | ✓ | ✓ | | systematic | 7K |
| CLEVR (Johnson et al., 2017) | synthetic | QA | i&t | ✓ | ✓ | | i.i.d | 865K |
| HWF (Li et al., 2020) | real | IC | image | ✓ | | | i.i.d | 12K |
| MNIST-Add (Manhaeve et al., 2018) | real | IC | image | ✓ | | | i.i.d | - |
| HINT (ours) | real | QA | image | ✓ | ✓ | ✓ | systematic | 1M |

## 2 RELATED WORK

**Benchmarks on Systematic Generalization** Although several benchmarks (Lake & Baroni, 2018; Hupkes et al., 2020; Barrett et al., 2018; Zhang et al., 2019; Teney et al., 2020; Keysers et al., 2020; Bahdanau et al., 2019; Ruis et al., 2020; Kim & Linzen, 2020; Keysers et al., 2020) have advanced systematic generalization, the majority of them are based on artificial domains with synthetic tasks, involve just one or two aspects of concept learning and often mixing the generalization over syntax and semantics. SCAN (Lake & Baroni, 2018) is tasked with translating a natural language command into a sequence of operations in a simplified navigation domain using only syntax and semantics. CLEVR (Johnson et al., 2017) requires parsing questions (syntax) and grounding visual objects (perception), although objects themselves lack functional semantics. We refer readers to Table 2 for detailed comparisons of related datasets.

In contrast, the proposed HINT benchmark stems from the area of arithmetic reasoning with real handwriting images (at the primitive level, rather than the expression level) and requires joint learning of perception, syntax, and semantics. The precise definitions and boundaries of these meanings in HINT permit to build test splits to evaluate the specific generalizations. Notably, HINT possesses more complex semantics, which eliminates the undesirable bias towards syntactic generalization present in earlier datasets. The task of the HINT benchmark is inspired by the HWF dataset (Li et al., 2020) but requires full-spectrum learning of perception, syntax, and semantics. By going beyond an i.i.d train/test split in Li et al. (2020), HINT focuses on examining systematic generalization across many aspects of concepts.

**Methods on Systematic Generalization** To capture systematic generalization, new training regimes (Lake, 2019; Andreas, 2020; Akyürek et al., 2020; Zhang et al., 2022) and model architectures (Dessì & Baroni, 2019; Russin et al., 2019; Csordás et al., 2021; Gordon et al., 2019; Bergen et al., 2021) have been developed. Russin et al. (2019), for instance, expand a seq2seq model by segregating syntactic and semantic information. Csordás et al. (2021) investigate a variety of Transformer configurations to enhance its systematic compositionality. Andreas (2020) and Akyürek et al. (2020) investigate data enhancement for compositional generalization.

In particular, several neural-symbolic methods with domain-specific designs (Chen et al., 2020; Nye et al., 2020; Liu et al., 2020) achieve near-perfect accuracy on prior systematic generalization datasets like SCAN (Lake & Baroni, 2018). However, these neural-symbolic methods introduce certain non-trivial domain-specific symbolic components, making it difficult to transfer to other domains; their flexibility and transferability are unclear. In this paper, we benchmark on HINT with prevailing seq2seq frameworks, including RNNs, Transformers, and GPT-3, which require minimal domain-specific design and may be of broad interest to the learning community. We reserve for future research the investigation of more sophisticated methods, such as data augmentation and neural-symbolic approaches.

## 3 THE HINT DATASET

In this section, we present the specifics of the HINT benchmark, devised to evaluate models' capability of learning generalizable concepts at three distinct levels: perception, syntax, and semantics.

## 3.1 The Definitions of Perception, Syntax, and Semantics

We first define the perception, syntax, and semantics in the domain of HINT, as shown in Table 3. *Perception* refers to the mapping from image pixels into meaningful patterns, such as mapping an image of handwritten expression to a symbolic sequence. *Syntax* refers to the mechanism of how the concepts in one sample are structurally organized *e.g.*, parsing the symbolic sequence into a tree, and the syntax in Table A2 is expressed by a phrase-structure grammar. *Semantics* refers to the functional meanings of these arithmetic concepts, *e.g.*, what value '5' represents and what value '+' produces when given two arguments 1 and 1.

Table 3: **The definitions of perception, syntax, and semantics** In syntax, *number*, *op1*, and *op2* are the HINT grammar's pre-terminals in Table A2. In semantics, $i$ and $j$ are the operator's inputs. $-$ is defined as $\max(0, i - j)$ to prevent negative results, and $\div$ is defined as $\text{ceil}(i \div j)$ to remove the decimal portions of the results.

(a) main concepts

| concept | perception | syntax | semantics |
|---------|-----------|--------|-----------|
| 0..5..9 | 0..5..9 | number | 0..5..9 |
| ( ) | ( ) | parenthesis | none |
| + | + | op1 | $i + j$ |
| − | − | op1 | $\max(0, i - j)$ |
| × | × | op2 | $i \times j$ |
| ÷ | ÷ | op2 | $\text{ceil}(i \div j)$ |

(b) new concepts in the few-shot learning split

| concept | perception | syntax | semantics |
|---------|-----------|--------|-----------|
| $x$ | $x$ | number | 11 |
| $y$ | $y$ | number | 12 |
| $a$ | $a$ | op1 | $\max(i, j)$ |
| $b$ | $b$ | op1 | $\min(i, j)$ |
| $c$ | $c$ | op2 | $(i + j) \div 2$ |
| $d$ | $d$ | op2 | $2i \times j \div (i + j)$ |

Notably, although these three levels have a clear boundary by their definitions, a model need not necessarily represent them by separate and individual modules. An end-to-end neural network trained on this domain, for instance, will likely contain neurons and parameters from all three layers. The notion of perception, syntax, and semantics simply requires the models to capture these meanings during evaluation, regardless of how the models finish the tasks, implicitly or explicitly.

**Task** The task of HINT is intuitive: predict the final results of handwritten arithmetic expressions in a weakly-supervised manner. That is, only the final results are given as supervision; all the symbolic expressions, parse trees, and intermediate values are latent. In such a setting, any model must simultaneously master perception, syntax, and semantics to solve this task successfully.

## 3.2 Data Generation

Table 4: **Examples from the training set and the test subsets of HINT.**

| Train | | $2 \times 5 \div 9$  2 $(9-9) \times (3-4) - 1 \times (0+3-(6-(9-2 \div 2)))$  0 |
| | | $5 \times 5 + (9-0 \sim 2)$  32  $4 \times (3+9)-7-(0-5)$  41 |
| Test | I | $1 \div 4$  1  $1 \times (2 \div 5) \times (8-8-6)$  0  $6-4+(0-(6+0 \div (4 \div (6/\land) \times 1))] + (9+4 \setminus 15$ |
| | SS | $1 + 3 \div 4$  2  $3 \times (7 \times 1) + (8+4) + 1 \times 3$  66  $4 + (0-(7+7+6)) \times 4 - 0$  4 |
| | LS | $9 \times (8 \times (8 \times 1) + 0 \div 9)$  192  $5 \times (3 \cdot 1 \times 9) + (2-5) \times (7 \times (6+5))$  135 |
| | | $2 \times (3 \times (3 \div 6 + 6 \times (3 \times 4 \times 6 \div (1 \times 6))) + 0 \div 3)$  **438** |
| | SL | $(6 \times 5 - 0) \div (1 + 9 + 5) \div 9) + (3-((2-(2+(9 \times 7-8 \div 9)))/1-9))$  18 |
| | | $6-3 \div (9 \times (9 \div (4-(4-7)))) + (1+1/1 b-1) \div (7 \times 1+6 \div 8)$  6 |
| | | $(7+3)/(6-1 \times (0 \times (6 \div 1))) - (3 \times 1-6-4/(4-3)) \times (9 \times 3)$  2 |
| | LL | $(6+2 \times 1+2 \div 4 + (1+4-0 \div 3) \times 8-(4+3 \times 8)) \times ((0+(2 \times 9-0)/3) \div (8 \div 9))$  174 |
| | | $(3+(8+(4-7 \times (7+8)) \times (8 \div 4-(4-(6+5)+6)))) \div (7+8 \times 1 \times 0) \div 5$  1 |
| | | $9 \times (8 \div (1 \times (7 \div 7)) + (1+2)) \times 10 + 9-5 \div (8+1 \div (9 \times 6))) + (8-(9-8+3))$  620 |

The data generation process consists of three steps. First, we extract handwritten images for each concept from CROHME (Mahdavi et al., 2019), including digits 0 through 9, operators $+, -, \times, \div$, and parentheses $(, )$. Second, we randomly sample *prefix* expressions and convert them to *infix* expressions with necessary parentheses based on the operator precedence; only single-digit numbers are permitted. The symbolic expressions are fed into a solver to calculate the final results. Third, we randomly sample handwritten images for symbols in an expression and concatenate them to construct the final handwritten expression. We only retain the handwritten expressions as input and the corresponding final results as supervision; all intermediate results are discarded.

**Full-Spectrum Systematic Generalization** To rigorously evaluate the systematic generalization of the learned concepts, we substitute the standard i.i.d. split with a meticulously crafted evaluation

scheme. We randomly divide all handwritten images into three splits: training (75%), validation (5%), and test (20%). First, we limit the maximum number of operators in the training set to 10 and the maximum intermediate values to 100:

$$D_{\text{train}} \subset \mathcal{T}_{\text{train}} = \{(x, y) : |x| \leqslant 10, \max(v) \leqslant 100\}, \tag{1}$$

where $x$ is the expression, $|x|$ its number of operators, $y$ the final result, and $v$ all the intermediate values and the final results. To ensure diversity in the training set, we sample a maximum of 100,000 distinct expressions with the same number of operators. To prevent bias in the final results, we cap the percentage of a certain result at less than 5%. Next, we carefully curate the test set to evaluate different generalization capabilities (*i.e.*, interpolation and extrapolation) on different levels of meaning (*i.e.*, perception, syntax, and semantics). Specifically, the test set comprises five subsets, formally defined as:

$$D_{\text{test}} = \text{I} \cup \text{SS} \cup \text{LS} \cup \text{SL} \cup \text{LL}, \text{where} \tag{2}$$

| | |
|---|---|
| $\text{I} \subset D_{\text{train}},$ | generalization on perception only |
| $\text{SS} \subset \mathcal{T}_{\text{train}} \backslash D_{\text{train}},$ | interpolation on both syntax and semantics |
| $\text{LS} \subset \{(x, y) : |x| > 10, \max(v) \leqslant 100\},$ | extrapolation on syntax and interpolation on semantics |
| $\text{SL} \subset \{(x, y) : |x| \leqslant 10, \max(v) > 100\},$ | interpolation on syntax and extrapolation on semantics |
| $\text{LL} \subset \{(x, y) : |x| > 10, \max(v) > 100\}.$ | extrapolation on both syntax and semantics |

All subsets of the test set require generalization on perception since all images in the test set are unseen during training. For the test set, we sample no more than 1,000 unique expressions with the same number of operators, and the final results are also balanced. The maximum number of operators is set up to 20, and the maximum intermediate value to 10,000. We also build a small validation set for hyperparameter tuning. See Table 4 for training and test examples and refer to Appendix A for further dataset statistics.

**Few-shot Learning and Generalization**   To determine if models can rapidly learn new concepts, we constructed a few-shot learning split to learn six new concepts, as shown in Table 3. These six concepts have different meanings in terms of perception, syntax, and semantics: two new numbers ($x$ 〻 and $y$ 〼 , representing 11 and 12, respectively), two operators of precedence 1 ($a$ 〇 and $b$ 〈, representing $\max$ and $\min$), and two operators of precedence 2 ($c$ 〈 and $d$ 〉, representing arithmetic mean and harmonic mean). The train, validation, and test splits are constructed using the same strategy as in the full-spectrum generalization. Expressions are sampled to guarantee that the corresponding new concept appears at least once in the expression. This few-shot learning split is used to determine whether the models pre-trained on the training set can rapidly learn a new concept by fine-tuning on only a handful of examples involving the new concept. In this context, "few-shot" implies that the examples used to acquire a new concept are significantly fewer than those of the training set, but still exceed the number of examples required by humans to learn a new concept.

## 4   DEEP SEQUENCE-TO-SEQUENCE BASELINES

The task of HINT can be naturally formulated as a sequence-to-sequence (seq2seq) problem: The input is a handwritten expression, segmented into a sequence of images by a sliding window, and the output is an integer, converted into a sequence of digits. We benchmark deep seq2seq frameworks on HINT; see Figure 1 for an illustration using a detailed example.

### 4.1   IMAGE TOKENIZING AND EMBEDDING

Existing seq2seq frameworks typically accept a sequence of tokens as input. To tokenize a handwritten expression, its height is first resized to 32, and a 32-pixel sliding window is applied along the horizontal axis to render a sequence of images. Next, each image in the sequence is encoded by ResNet-18 (He et al., 2016), sufficient to handle the visual variance in handwriting.

### 4.2   ENCODER-DECODER ARCHITECTURES

**RNNs**   Recurrent neural networks (RNNs) have long been a dominant choice for sequence modeling tasks. We test two popular RNNs in the literature: long short-term memory (LSTM) (Hochreiter & Schmidhuber, 1997) and gated recurrent units (GRU) (Chung et al., 2014). Each model is evaluated both with and without attention (Bahdanau et al., 2015).

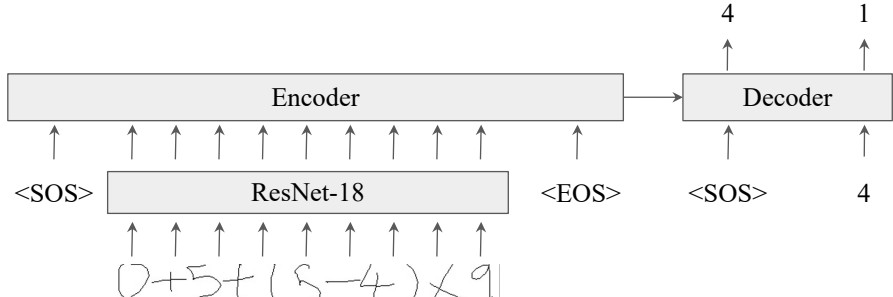

Figure 1: **The seq2seq framework applied to an example in HINT.** ¡SOS¿: start-of-sentence tokens. ¡EOS¿: end-of-sentence tokens. A sliding window segments the handwritten expression into a sequence of images, which are then separately encoded by ResNet-18. The expected output is a sequence of digits in reverse order.

**Transformers**   Since its inception (Vaswani et al., 2017), Transformers have gradually supplanted recurrent or convolutional neural networks as the *de facto* choice for various sequence modeling tasks (Devlin et al., 2019; Radford et al., 2019; Brown et al., 2020). Nevertheless, prior work (Dehghani et al., 2018; Hupkes et al., 2020; Kim & Linzen, 2020) suggests that the vanilla Transformer fails substantially in many tasks requiring systematic generalization when the sequence lengths exceed those observed during training. Recently, several simple tricks have been proposed (Csordás et al., 2021) to improve the generalization capability of Transformers; two of them work particularly well: (i) using relative positional encoding (Shaw et al., 2018; Dai et al., 2019), and (ii) sharing weights across the blocks in the Transformer, *a.k.a.*, Universal Transformer (Dehghani et al., 2018). Therefore, we benchmark Transformer variants: the vanilla Transformer, Transformer with relative positional encoding, and Universal Transformer with relative positional encoding.

**GPT-3**   Since the commencement of GPT-3 (Brown et al., 2020), there have been intense debates and different perspectives regarding the mathematical reasoning capacity of pre-trained large language models.[1] To systematically and comprehensively evaluate GPT-3's competence of arithmetic reasoning, we test it on the proposed HINT benchmark using symbolic expressions as input. Since all tokens of HINT are in the vocabulary of GPT-3, we directly evaluate GPT-3 via zero-shot prompting using the OpenAI API. [2] We construct the prompt in the following form: "*Q: What is Expression? A: The answer is*", similar to the practice in Brown et al. (2020), but with more complex expressions.

Recently, chain of thought (CoT) prompting (Wei et al., 2022) has been extended to the zero-shot setting (Kojima et al., 2022) by adding a simple prompt, "*Let's think step by step,*" to facilitate step-by-step thinking prior to answering each question. Zero-shot CoT surpasses the standard zero-shot prompting by a significant margin in various reasoning tasks. Therefore, we also apply zero-shot CoT prompting to evaluate GPT-3 on HINT; we refer the readers to Appendix B.2 for the details of zero-shot CoT.

### 4.3   TRAINING AND EVALUATION

**Training**   All models are trained using the Adam optimizer (Kingma & Ba, 2014); the gradients exceeding 5.0 are clipped. Dropout (Srivastava et al., 2014) is applied to each recurrent layer of RNNs and each sub-layer of Transformers, including both the multi-head attention layers and the feedforward layers. No training is required for zero-shot experiments on GPT-3; instead, 100 samples from each test subset are selected and fed to GPT-3 through zero-shot or zero-shot-CoT prompting.

**Hyperparameter Tuning**   To produce reliable results, a thorough hyperparameter tuning is performed w.r.t. the number of layers in the encoder and the decoder, the dimension of the token embedding, the number of hidden units per layer, the number of attention heads in Transformers, the dropout ratio, and the learning rate. We refer the readers to Table A3 for further information.

**Evaluation Metric**   We report the accuracy of the final results. A predicted result is considered correct only when it *exactly* matches the ground-truth answer.

---

[1]Can GPT-3 do math? `https://www.youtube.com/watch?v=TMxAbNAVrzI`
[2]`https://openai.com/api/`

Table 5: **The accuracy on the test set using image inputs.** All models are jointly trained with a randomly initialized ResNet-18. Reported accuracy (%) is the median and standard deviation of 5 runs. "rel." denotes Transformer with relative positional encoding, and "uni." denotes Universal Transformer.

| Model | Variant | I | SS | LS | SL | LL | Avg. |
|---|---|---|---|---|---|---|---|
| GRU | w/o att | 61.3±1.4 | 53.3±1.7 | 30.5±1.2 | 9.2±0.2 | 11.9±0.5 | 33.2±0.9 |
| | w/ att | 66.7±2.0 | 58.7±2.2 | 33.1±2.7 | 9.4±0.3 | 12.8±1.0 | 35.9±1.6 |
| LSTM | w/o att | 80.0±5.7 | 76.2±7.4 | 55.7±8.2 | 10.9±0.6 | 19.8±2.6 | 48.6±4.9 |
| | w/ att | 83.9±0.9 | 79.7±0.8 | 62.0±2.5 | **11.2±0.1** | **21.0±0.8** | 51.5±1.0 |
| Transformer | vanilla | 20.9±0.4 | 9.3±0.2 | 5.7±0.3 | 1.5±0.3 | 2.9±0.5 | 8.3±0.3 |
| | rel. | 86.2±0.9 | 83.1±1.3 | 60.1±2.3 | 10.9±0.2 | 19.4±0.5 | 51.7±1.0 |
| | rel. uni. | **88.4±1.3** | **86.0±1.3** | **62.5±4.1** | 10.9±0.2 | 19.0±1.0 | **53.1±1.6** |

Table 6: **The accuracy on the test set using symbol inputs.**

| Model | Variant | I | SS | LS | SL | LL | Avg. |
|---|---|---|---|---|---|---|---|
| GRU | w/o att | 74.9±1.6 | 68.1±0.5 | 42.1±1.9 | 10.5±0.2 | 14.0±0.8 | 41.3±0.6 |
| | w/ att | 76.2±0.6 | 69.5±0.6 | 42.8±1.5 | 10.5±0.2 | 15.1±1.2 | 42.5±0.7 |
| LSTM | w/o att | 84.3±5.2 | 79.6±6.0 | 63.7±6.1 | 11.7±0.3 | 22.1±1.4 | 52.3±3.8 |
| | w/ att | 92.9±1.4 | 90.9±1.1 | 74.9±1.5 | **12.1±0.2** | **24.3±0.3** | 58.9±0.7 |
| Transformer | vanilla | 93.9±0.3 | 91.0±0.5 | 33.2±1.2 | 11.5±0.1 | 11.5±0.7 | 47.4±0.4 |
| | rel. | 96.6±0.3 | 95.1±0.4 | 72.1±1.5 | 11.8±0.2 | 22.3±0.6 | 59.4±0.5 |
| | rel. uni. | **98.0±0.3** | **96.8±0.6** | **78.2±2.9** | 11.7±0.3 | 22.4±1.1 | **61.5±0.9** |
| GPT-3 | 0-shot | 19.0 | 9.0 | 3.0 | 10.0 | 2.0 | 8.6 |
| | 0-CoT | 42.0 | 36.0 | 5.0 | **49.0** | 6.0 | 27.6 |

## 5 RESULTS

### 5.1 JOINT LEARNING OF PERCEPTION, SYNTAX, AND SEMANTICS

Tables 5 and 6 summarize the results of all models on HINT using image inputs and symbol inputs, respectively. Among all models, the Universal Transformer with relative positional encoding ("Transformer rel. uni.") has the highest average accuracy on the test set. Upon careful examination of the results, the following observations and insights can be made:

- **Models attains high accuracy on the subset I.** Particularly, Transformer rel. uni. using image inputs achieves an accuracy of 88.4%. The test subset I shares the symbolic expressions with training and has different handwritten images for symbols. This indicates that Transformers and RNNs, jointly trained with ResNet-18, have strong generalization over perception. As depicted in Figure 2, the model forms meaningful clusters for each concept and captures syntactic roles to some extent without direct supervision on perception.

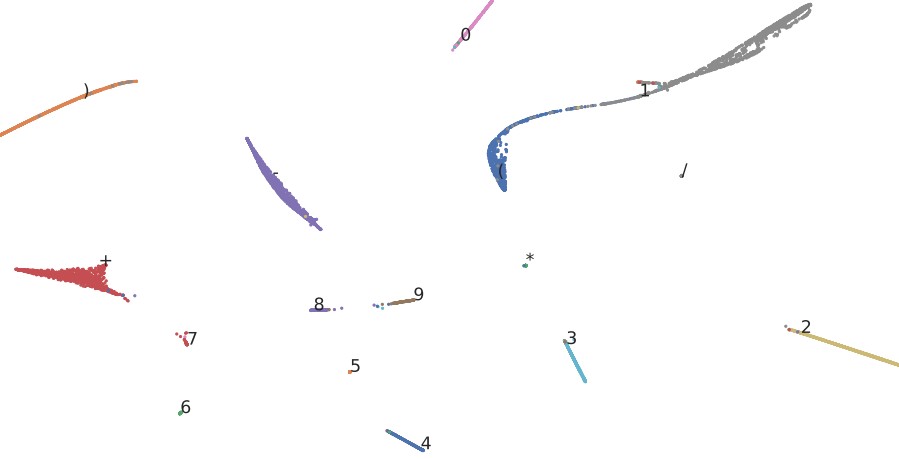

Figure 2: **The t-SNE visualization of the embeddings (the outputs of ResNet-18) of handwritten images using the Transformer rel. univ. model.** The image embeddings form clear clusters for each concept based on visual appearance. In addition, these clusters reflect the concepts' syntactic roles: The majority of digits are towards the bottom, operators are around the center, and parentheses are near the top.

- **Transformers achieve high accuracy on the subset SS.** The expressions in SS share the same length and value distribution as training. This result indicates that Transformers exhibit robust interpolation over syntax and semantics.

- **The accuracy of Transformer rel. uni. on LS is substantially lower than its accuracy on SS or I** (see Table 6). Note that the identical model yields perfect accuracy on the length cutoff splits of SCAN (Csordás et al., 2021). This result, however rather unexpected, may be explained by the syntax difference between HINT and SCAN shown in Table A2: The expressions in HINT may have a longer-range dependency and greater tree depth than the commands in SCAN. This observation suggests that present Transformers, which have finite depth, are incapable of adequately capturing the syntax with long dependencies and large depth.

- **Transformer with relative positional encoding achieves similar performance on I and SS as the vanilla Transformer with absolute positional encoding, yet relative positional encoding doubles the Transformer's accuracy on LS** (see Table 6). This contradiction implies that relative positional encoding is essential for Transformer to generalize to long expressions. Sharing weights between the layers using the Universal Transformer can further enhance performance.

- **Models behave clumsily on the subsets SL and LL.** The accuracy on SL and LL is significantly lower than that on I and SS. All models exhibit near-zero accuracy on samples whose answers are over 100 (the maximum final result in the training set). This finding suggests that neither RNNs nor Transformers are able to extrapolate to larger numbers beyond those in the training set.

- **While GPT-3 with zero-shot prompting performs poorly, chain of thought (CoT) prompting significantly improves the accuracy.** Notably, GPT-3 with zero-shot CoT achieves an accuracy of 49.0% on SL, which is superior to other fine-tuned models. We believe this is due to the fact that GPT-3 has been pre-trained on data with larger numbers, and CoT improves the reasoning process. Despite CoT prompting, GPT-3 performs poorly on long expressions in LS and LL.

**Summary** We observe a significant room for improvement on HINT. Even the best model, Universal Transformer with relative positional encoding, can only achieve an accuracy of 54.3% on HINT, while the same model achieves virtually perfect accuracy on earlier datasets of systematic generalization, such as SCAN. The challenges of HINT stem from the fact that it requires joint learning and generalization of perception, syntax, and semantics: The perception has a large variance in real handwriting, the syntax supports long dependency between symbols, and the semantic complexity is well beyond the capability of the state-of-the-art models.

**Scaling Laws** Since HINT can generate an endless amount of data for training, one may wonder if merely increasing the dataset and the model size can solve the problem, akin to certain NLP tasks (Kaplan et al., 2020; Henighan et al., 2020). Empirically, Figure 3 depicts the test accuracy's scaling trend w.r.t. the model size and the number of training samples. By altering the hidden dimensions, the embedding dimension, and the number of attention heads, various-sized models are constructed. Similarly, various-sized training sets are generated by randomly sampling the original training set. Assuming a log-linear scaling trend, we need to train a model of $10^{33}$ parameters on $10^{15}$ examples to attain 90% accuracy on the test subset LL, which is impractical. Hence, efficient architectures and training algorithms are still in need to improve extrapolation over syntax and semantics.

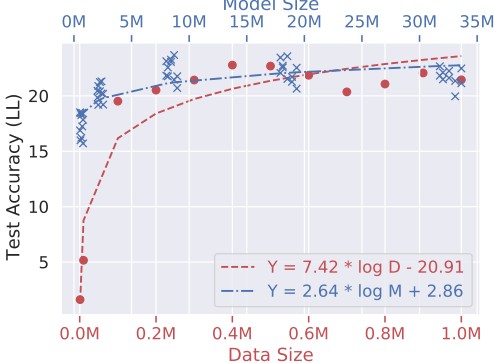

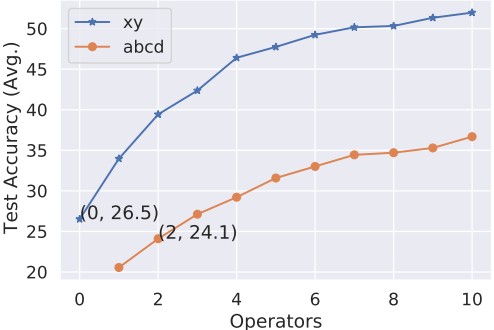

Figure 3: **Scaling trends w.r.t. model size and dataset size** when training Transformer rel. uni. on the test subset LL with symbol inputs.

Figure 4: **The few-shot learning performance when training Transformer rel. uni. with varied maximum operators.**

## 5.2 FEW-SHOT LEARNING AND GENERALIZATION

In this section, we fine-tune the top two models on six new concepts; Table 7 summarizes the results. Transformer rel. uni. outperforms LSTM w/ attn across all concepts by a significant margin, which is greater than six times their performance gap in Table 5. This discrepancy suggests that with limited data, Transformer is superior to LSTM at learning new concepts.

Table 7: **The few-shot learning performance of the top two models: LSTM w/ attn (left) and Transformer rel. uni. (right).** Reported results are the median of 5 runs. See Table 3 for the meanings of these concepts. *Please refer to Appendix C for the details regarding the human study.

| Concept | I | SS | LS | SL | LL | Avg. | Human* |
|---------|---|----|----|----|----|------|--------|
| $x$ | 87.8/89.2 | 47.3/80.2 | 42.8/58.6 | 10.8/12.2 | 16.4/19.3 | 42.8/52.8 | 95.0 |
| $y$ | 64.5/83.8 | 39.1/74.8 | 38.5/54.0 | 11.6/13.8 | 18.9/22.4 | 35.4/50.7 | 100.0 |
| $a$ | 71.8/84.4 | 44.2/72.0 | 29.7/48.9 | 7.9/8.4 | 11.1/12.3 | 33.8/46.4 | 97.5 |
| $b$ | 73.4/77.1 | 29.9/59.1 | 27.4/39.4 | 7.4/16.8 | 12.7/17.1 | 31.1/42.6 | 77.5 |
| $c$ | 61.5/59.2 | 19.6/34.0 | 15.2/24.4 | 4.5/6.1 | 6.5/9.4 | 21.9/27.3 | 90.0 |
| $d$ | 59.2/62.8 | 22.7/39.0 | 20.2/27.0 | 7.2/9.2 | 8.9/10.7 | 24.7/30.4 | 60.0 |
| Overall | 69.7/76.1 | 33.8/59.9 | 29.0/42.0 | 8.2/11.1 | 12.4/15.2 | 31.6/41.7 | 86.7 |

Figure 4 depicts the test accuracy of Transformer rel. uni. while using varied maximum operators for training. In general, the more data and longer expressions used for training, the higher the model's performance. One test case for learning new numbers ("$xy$") is $(0, 26.5)$, where the model is only exposed to the primitive concept during training and is expected to generalize to complex compositions during testing. The classic thought experiments (Fodor, 1975) indicate that this is straightforward for humans: If you grasp the meanings of "1," "$1 + 1$," and "$x$," you should also comprehend the meaning of "$1 + x$". A similar test case for learning new operators ("$abcd$") is $(2, 24.1)$ since expressions comprising at least two operators are required to capture the syntax of a new operator. Transformer performs poorly on both of these tasks, demonstrating that it is still far from human-level generalization.

## 6 DISCUSSIONS: CONCLUSIONS AND LIMITATIONS

In this paper, we took inspiration from arithmetic and introduced a new challenge for the learning community, Handwritten arithmetic with INTegers (HINT), which serves as a minimal yet comprehensive benchmark for examining the full-spectrum systematic generalization of concept learning w.r.t. perception, syntax, and semantics. HINT is intrinsically more challenging than previous datasets on systematic generalization due to its substantial perceptual diversity in real handwriting, complex syntax, and sophisticated semantics. We benchmark on HINT with the state-of-the-art seq2seq models, including RNNs, Transformers, and GPT-3; the results point out their inability to extrapolate over syntax and semantics. The scaling trends of test accuracy w.r.t. dataset size and model size indicate that it is impractical to solve HINT by only increasing the size of the dataset and model. We believe that the HINT dataset and our experimental findings will inspire new advances in systematic generalization, particularly extrapolation over syntax and semantics.

**Limitations and Future Work** Despite a large visual variance, the handwritten expressions are rather basic in terms of spatial locations and visual complexity. It would be more intriguing if we could further increase the perceptual complexity w.r.t. spatial relations like natural images (Lin et al., 2014). Although syntax and semantics in HINT are already more complex than those of prior datasets, they remain context-free. Extending our findings to context-dependent syntax and semantics would be of practical value given their prevalence in natural languages; *e.g.*, a word might have different syntactic roles or semantic meanings in different contexts.

Regarding model development on HINT, our findings reveal that current seq2seq models, including Transformers, are unable to extract the systematic rules for both syntax and semantics from the training data. Improving the systematic generalization of Transformers, particularly extrapolation over semantics, is a crucial future direction. We also intend to investigate more advanced methods, such as meta-learning (Lake, 2019), data augmentation (Andreas, 2020; Akyürek et al., 2020), Edge Transformer (Bergen et al., 2021), and Neural-Symbolic Stack Machines (Chen et al., 2020). In addition, understanding the systematic generalization of large language models by evaluating them in few-shot or fine-tuning settings will be beneficial.

**Acknowledgements.** The authors would like to thank four anonymous reviews for constructive feedback. This work is supported in part by the National Key R&D Program of China (2021ZD0150200) and the Beijing Nova Program.

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

## A  DATASET STATISTICS

The handwritten images for each arithmetic concept originate from the handwritten math symbols dataset[1] hosted on Kaggle under the "CC0: Public Domain" license, parsed and extracted from the Competition on Recognition of Online Handwritten Mathematical Expressions (CROHME) (Mahdavi et al., 2019)[2]. We further clean the dataset by removing duplicate images, resulting in statistics shown in Figure A1.

We conduct a detailed analysis of the collected data to demonstrate the validity of the HINT dataset as a benchmark for systematic generalization. Table A1 shows the size of each split in HINT, and Table A2 shows a comparison between the grammars of HINT and SCAN.

---

[1] https://www.kaggle.com/datasets/xainano/handwrittenmathsymbols
[2] https://www.cs.rit.edu/~crohme2019/

Table A1: **Dataset size.** The first row is the main split of HINT, and the rest are the few-shot learning split. As advocated by Csordás et al. (2021), the validation set also contains five generalization subsets for model selection.

| Split | Train | Validation | Test | | | | | |
|---|---|---|---|---|---|---|---|---|
| | | | Total | I | SS | LS | SL | LL |
| main | 998000 | 4698 | 46620 | 9980 | 8000 | 10000 | 8640 | 10000 |
| $x$ | 1100 | 491 | 4900 | 1100 | 900 | 1000 | 900 | 1000 |
| $y$ | 1100 | 493 | 4900 | 1100 | 900 | 1000 | 900 | 1000 |
| $a$ | 1000 | 470 | 4700 | 1000 | 900 | 1000 | 800 | 1000 |
| $b$ | 1000 | 470 | 4700 | 1000 | 900 | 1000 | 800 | 1000 |
| $c$ | 1000 | 470 | 4700 | 1000 | 900 | 1000 | 800 | 1000 |
| $d$ | 1000 | 470 | 4700 | 1000 | 900 | 1000 | 800 | 1000 |

Table A2: **The phrase-structure grammars for HINT and SCAN.** While the grammars of both HINT and SCAN can generate infinite examples, HINT produces examples with larger depth and longer dependency due to the parentheses; the expression inside parentheses can be arbitrarily long. Specifically, the maximum depth and dependency range in SCAN are 6 and 4, respectively; the maximum length generated by the non-terminal "S" in the grammar of SCAN is 4.

| HINT | SCAN |
|---|---|
| T = {Expression, Term, Factor, Number} | T = {C, S, V, D, U} |
| Start symbol: Expression | Start symbol: C |
| $\Sigma = \{+, -, \times, \div, 0, 1, ..., 9, (, )\}$ | $\Sigma = \{$walk, look, run, jump, turn, left, right, |
| R = { | around, opposite, and after, twice, thrice$\}$ |
| $\quad$ Expression → Term | R = { |
| $\quad$ Expression → Expression Op1 Term | $\quad$ C → S — S and S — S after S |
| $\quad$ Op1 → + $\mid$ − | $\quad$ S → V — V twice — V thrice |
| $\quad$ Term → Factor | $\quad$ V → D[1] opposite D[2] |
| $\quad$ Term → Term Op2 Factor | $\quad$ V → D[1] around D[2] |
| $\quad$ Op2 → × $\mid$ ÷ | $\quad$ V → D — U |
| $\quad$ Factor → Number | $\quad$ D → U left — U right |
| $\quad$ Factor → ( Expression ) | $\quad$ D → turn left — turn right |
| $\quad$ Number → $0\mid1\mid2\mid3...\mid9$ } | $\quad$ U → walk — look — run — jump } |

For each split, we plot the frequency distributions of various aspects, including symbol, number of operators, expression length, tree depth, maximum dependency range, and result, as shown in Figure A2. The symbol distributions are similar across different splits, and the Kullback–Leibler divergence between train and test is low (0.0055). The digits and operators are approximately equally distributed, except for the test-SL split. The test-SL split has a relatively higher portion of multiplication ('*') since generating large numbers generally requires more multiplication for short expressions.

The test set's result distributions differ from the train set. All results in the training set are smaller than 100 as desired; about half are in $[0, 10)$. In comparison, 29% of the results in the test set are larger than 100.

Several properties of an input expression, including length, number of operators, tree depth, and maximum dependency range, are indicators of the difficulty of calculating the expression. We plot the frequency distributions w.r.t. these input properties in Figure A2. These distributions demonstrate significant differences between train and test.

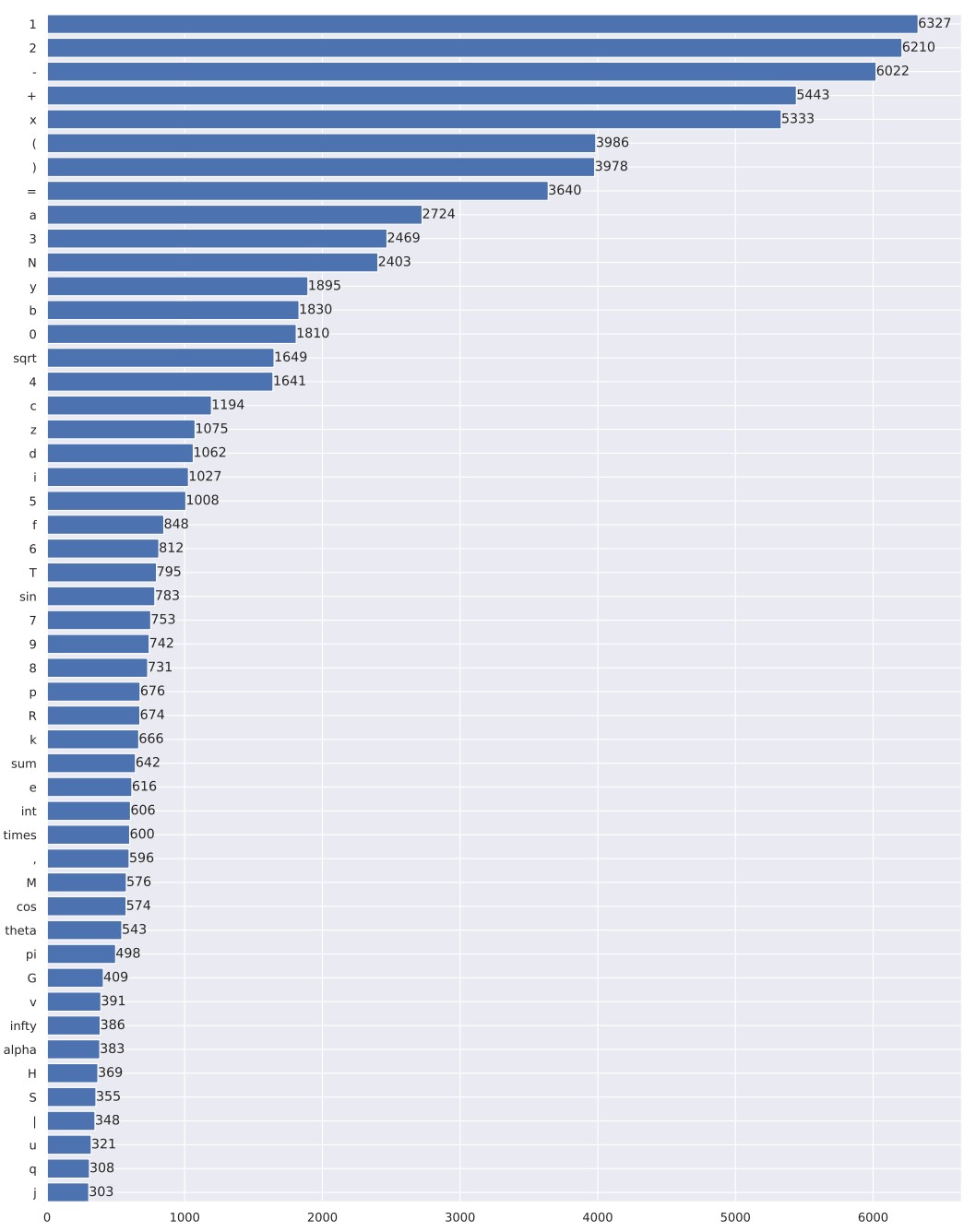

Figure A1: **The number of handwritten images for each symbol.** There are 82 arithmetic symbols (the top 50 are shown here) and 83,501 images in total. We use the handwritten images for digits $0 \sim 9$, operators $+, -, \times, \div$, and parentheses $(,)$ in this work; others are for potential future use.

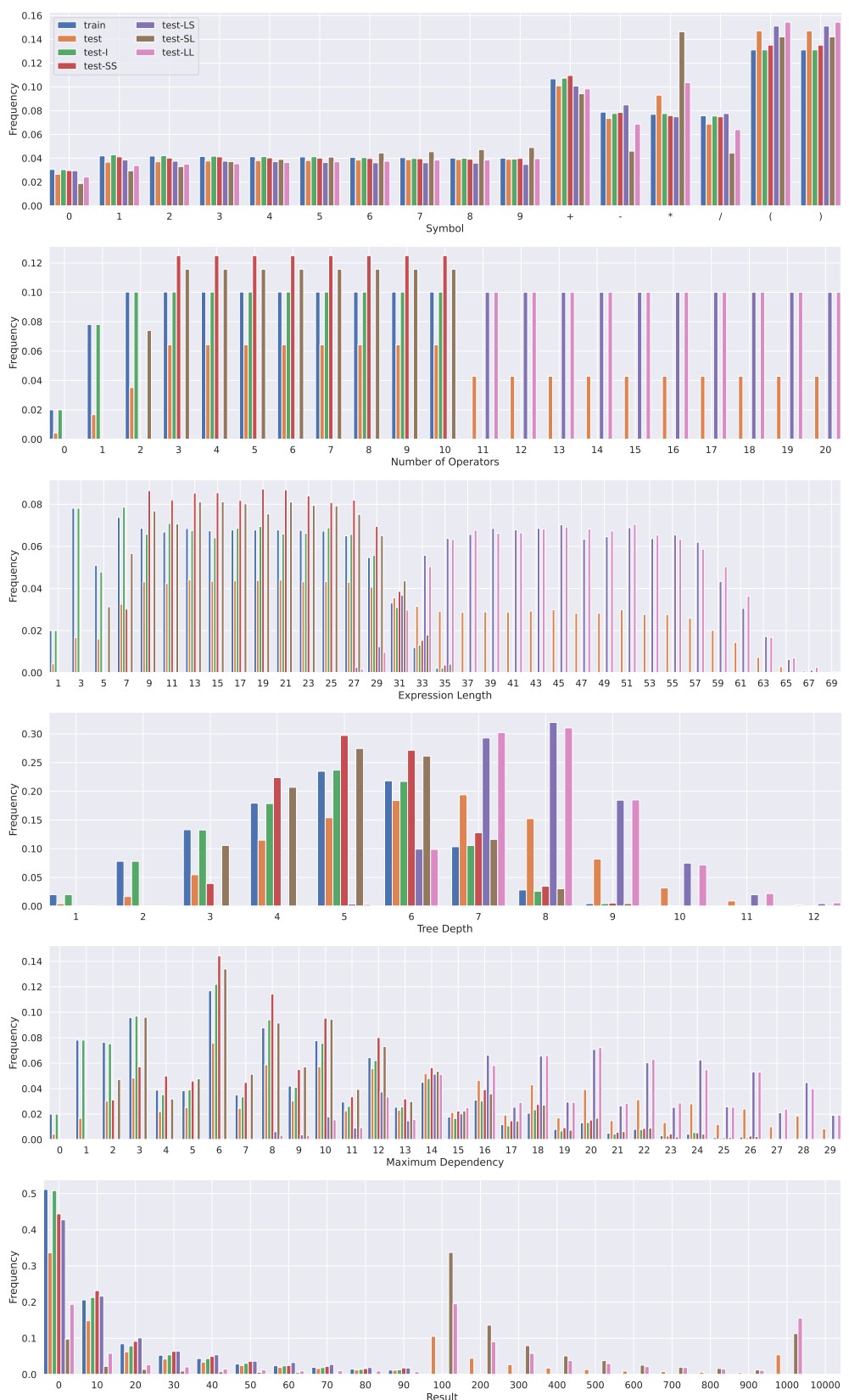

Figure A2: **The frequency distributions w.r.t. various aspects**, including symbol, number of operators, expression length, tree depth, maximum dependency range, and result.

# B  IMPLEMENTATION DETAILS

We benchmark deep sequence-to-sequence (seq2seq) frameworks on HINT, as illustrated by Figure 1. All models are implemented in PyTorch (Paszke et al., 2019).

## B.1  IMAGE TOKENIZER AND EMBEDDING

To tokenize a handwritten expression, we first resize it by making its height 32 and apply a sliding window of size 32 along the horizontal axis to render a sequence of images. Next, each image in the sequence is encoded by the ResNet-18 (He et al., 2016). We found in preliminary experiments that pre-training on the ImageNet does not help, likely due to the domain gap between ImageNet and HINT. Therefore, we use a random initialization for ResNet-18 in our experiments.

## B.2  ENCODER-DECODER ARCHITECTURES

We consider the following three choices for the encoder-decoder architecture in a seq2seq framework: Recurrent Neural Networks (RNNs), Transformers, and GPT-3.

**RNNs**  We test two popular RNNs: long short-term memory (LSTM) (Hochreiter & Schmidhuber, 1997) and gated recurrent units (GRU) (Chung et al., 2014). Both networks are evaluated with and without attention (Bahdanau et al., 2015). Our implementations of RNNs are adapted from a seq2seq tutorial.[3]

**Transformers**  We benchmark three variants of Transformer: the vanilla Transformer, Transformer with relative positional encoding, and Universal Transformer with relative positional encoding. The implementations of these Transformers are adapted from Csordás et al. (2021).[4]

**GPT-3**  To test GPT-3's ability to perform simple arithmetic operations without task-specific training, Brown et al. (2020) developed a small battery of 10 tests that involve asking GPT-3 a simple arithmetic problem in natural language; see Section 3.9.1 and Table 3.9 in Brown et al. (2020) for the results. In these tests, GPT-3 displays reasonable proficiency at simple arithmetic in the few-shot setting. However, they do not evaluate the multi-hop reasoning capability required by complex arithmetic expressions, which usually involve more operators and larger numbers.

To systematically and comprehensively evaluate GPT-3's capability of arithmetic reasoning, we test GPT-3 on the proposed HINT benchmark using symbolic expressions as input. Since all tokens of HINT are in the vocabulary of GPT-3, we directly evaluate GPT-3 via zero-shot prompting using the OpenAI API [5]. We construct the prompt in the following form: "`Q: What is <Expression>? A: The answer is`," similar to the practice in Brown et al. (2020) but with more complex expressions.

Via task-specific zero-shot or few-shot prompting, pre-trained large language models achieve excellent performance in intuitive and single-step *System 1* tasks Kahneman (2011). However, LLMs struggled on *System 2* tasks that require slow thinking and multi-hop reasoning (Rae et al., 2021), even at the scale of over 100B parameters like GPT-3. To address this shortcoming, *chain of thought* prompting (CoT) (Wei et al., 2022), which feeds LLMs with the intermediate step-by-step reasoning to augment the final answer in a few-shot setting, has been proposed to elicit the multi-hop reasoning in LLMs.

Very recently, chain of thought prompting has been extended to the zero-shot setting (Kojima et al., 2022) by adding a simple prompt, "`Let's think step by step`", to facilitate step-by-step thinking before answering each question. Zero-shot CoT amazingly outperforms the standard zero-shot prompting by a large margin in a variety of reasoning tasks. Therefore, we also apply zero-shot CoT prompting to evaluate GPT-3 on HINT. More concretely, it follows a two-stage prompting strategy similar to Kojima et al. (2022):

**1st prompt** "`Q: What is <Expression>? A: Let's think step-by-step.`" This prompt extracts the step-by-step reasoning process in the form of natural language from GPT-3,

---

[3] `https://github.com/bentrevett/pytorch-seq2seq`
[4] `https://github.com/RobertCsordas/transformer_generalization`
[5] `https://openai.com/api/`

Table A3: **Hyperparameter tuning.** Our choices are underlined.

| Model | Variant | Encoder | Decoder | Embedding | Hidden | Heads | Dropout | Batch | Steps | Learning Rate |
|---|---|---|---|---|---|---|---|---|---|---|
| RNN | LSTM (+ att) GRU (+ att) | 1,3,6,9 | 1,3,6,9 | 128, 256, 512 | 128, 256, 512 | - | 0, 0.1, 0.5 | 128 | 100K | $10^{-3}, 10^{-4}, 10^{-5}$ |
| Transformer | vanilla relative relative universal | 1,3,6,9 | 1,3,6,9 | 128, 256, 512 | 128, 256, 512 | 4,8,12 | 0, 0.1, 0.5 | 128 | 100K | $10^{-3}, 10^{-4}, 10^{-5}$ |

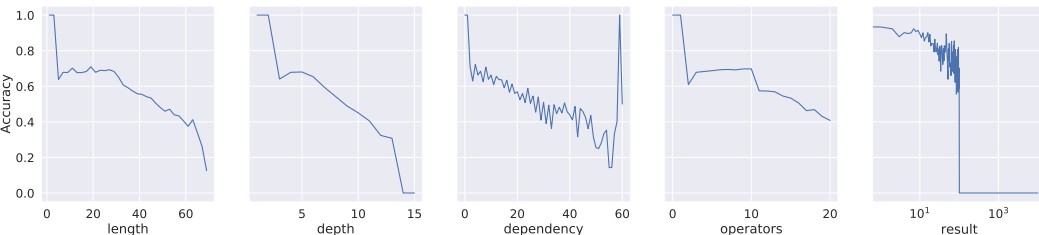

Figure A3: **Test accuracy (avg.) of Transformer rel. uni. using symbol inputs as a function of several properties of samples:** the expression's *length*, the *depth* of the expression's parse tree, the expression's maximum *dependency* range, the number of *operators* in the expression, the final *result*.

which is denoted by ¡Z¿.
**2st prompt** "Q: What is <Expression>? A: Let's think step-by-step. <Z> Therefore, the answer (arabic numerals) is" In the second stage, the response ¡Z¿ generated in the first step is appended to the initial prompt along with an answer trigger sentence. This second prompt is then fed into GPT-3 to predict the final answer.

In our experiments, we use the 'text-davinci-002' engine in the OpenAI API, the most capable GPT-3 model at the time of writing with approximately 175 billion parameters[6].

## B.3 TRAINING

Table A3 shows the tuned hyperparameters for the baselines. Our choices for each model are underlined, and the performance is reported under these settings unless explicitly stated otherwise. When generating the output, we use greedy decoding in all models for simplicity.

For the few-shot learning experiments, models are first pre-trained on the main training set and then fine-tuned on the training set of each new concept individually. Models are fine-tuned for 1000 iterations using a batch size of 128 with half examples from the main training set to prevent forgetting. The learning rates are $10^{-5}$ and $10^{-3}$ for Transformers and RNNs, respectively.

All models reported in our paper can be trained on a single NVIDIA TITAN V GPU with 12G memory. It takes at most eight hours to train a model.

## B.4 ADDITIONAL EXPERIMENTAL RESULTS

Figure A3 shows the test accuracy as a function of several sample properties. Figure A4 shows the importance of these properties.

## C HUMAN STUDY FOR FEW-SHOT LEARNING AND GENERALIZATION

We conduct a preliminary human study to evaluate human performance in the few-shot learning experiment. Specifically, we test ten human subjects on the six concepts that are unknown to subjects to reduce the human prior as much as possible. The human subjects are asked to determine each concept's meaning from 10 training examples and answer 4 test questions. We report the accuracy of test questions as human performance.

---

[6]OpenAI API GPT-3 model sizes: https://blog.eleuther.ai/gpt3-model-sizes

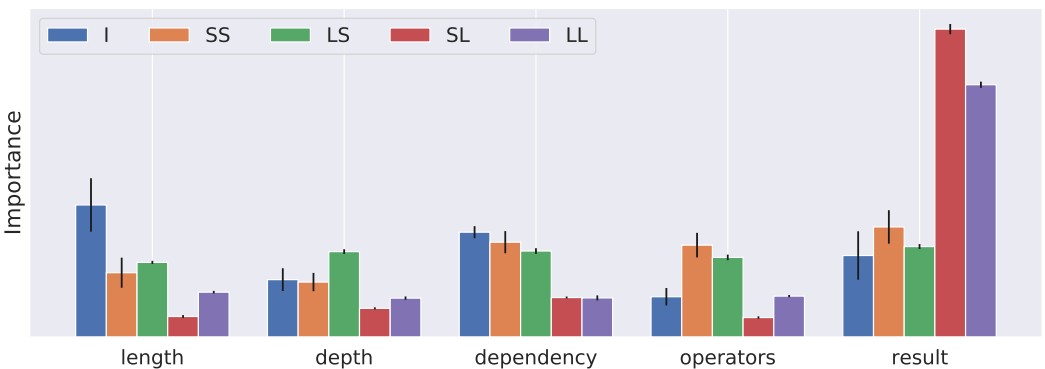

Figure A4: **The importance of sample properties w.r.t. the test accuracy of Transformer rel. uni. using symbol inputs.** Normalized permutation feature importance is reported here using a k-nearest neighbors classifier (k=3) to predict if the model can generate correct results.

