# OpenReview forum: "A Minimalist Dataset for Systematic Generalization of Perception, Syntax, and Semantics"
_ICLR.cc/2023/Conference — ICLR 2023 notable top 25%_

### Official Review · Reviewer_q55G · 2022-10-13

**Confidence:** 4
**Clarity, Quality, Novelty And Reproducibility:** The paper is very well written.
**Correctness:** 4
**Technical Novelty And Significance:** 3
**Empirical Novelty And Significance:** 2
**Recommendation:** 8

**Strength And Weaknesses:**

### Strengths

- Complementary to existing benchmarks(e.g. SCAN): realistic (non-synthetic) visual input, complex syntax and semantics.
- Experiments with a broad range of models (classical seq2seq to modern transformers).
- Various evaluation settings including a few-shot learning scenario.

### Weaknesses / suggestions

- W1: The existing datasets included in the review (Table 2) focus on those with training/test splits specifically designed to evaluate systematic/compositional generalization. It seems to me that other benchmarks for visual reasoning are also relevant to most other aspects of this work, such as the Raven-type matrices, for example [1,2,3]. [1,2] require perception and semantics (I believe), while [3] additionally requires perception of real visual inputs (entire photographs). I think this extension (perception requiring to parse entire photographs) is particularly relevant to this work (w.r.t. the first limitation - discussed by the authors - of the limited visual complexity of digits).

- W2: The chosen task/domain has several limitations, but the authors already discuss these quite clearly at the very end of the paper.


[1] [Measuring abstract reasoning in neural networks](https://arxiv.org/abs/1807.04225)

[2] [RAVEN: A Dataset for Relational and Analogical Visual rEasoNing](https://arxiv.org/abs/1903.02741)

[3] [V-PROM: A Benchmark for Visual Reasoning Using Visual Progressive Matrices](https://arxiv.org/abs/1907.12271)


**Summary Of The Paper:**

The paper introduces a visual reasoning task/benchmark that combines 3 levels: perception, syntax, and semantics. The reasoning aspect requires compositional generalization. Concretely, the task is to interpret handwritten arithmetic sequences, directly from images. Models are expected to learn all 3 levels end-to-end, without intermediate supervision. The evaluation is set up with different test set to measure performance of the various levels and with varying difficulty.

**Summary Of The Review:**

I think this task/benchmark is at the right level of complexity for the current state of the art. I cannot see any major flaws in the paper and believe it will be of interest to the research community.

---

> ### Author Response · Authors · 2022-11-19
> **Thank you for the constructive feedback!**
>
> We very much appreciate your efforts in reviewing our paper and providing insightful feedback. We are very encouraged to see your positive comments: "Complementary to existing benchmarks," "Experiments with a broad range of models," "Various evaluation settings," and "The paper is very well written."
>
> We address your concerns as follows:
>
> > Other benchmarks for visual reasoning are also relevant to most other aspects of this work
>
> Thank you for the pointers and helpful explanations! These three datasets are indeed related to our work in a broad sense. We will add these datasets to Table 2 for comparison in the revised draft.

---

### Official Review · Reviewer_LVqC · 2022-10-22

**Confidence:** 2
**Clarity, Quality, Novelty And Reproducibility:** Very nice across these aspects.
**Correctness:** 3
**Technical Novelty And Significance:** 3
**Empirical Novelty And Significance:** 3
**Recommendation:** 8

**Strength And Weaknesses:**

I want to preface this by saying that I'm not really familiar with this field.

Strength:
* The dataset is quite interesting to me. I think it can be a well-received dataset to test machine's ability to learn concepts.
* The presentation is quite clear. The authors show several very informative tables (e.g. comparison with prior datasets, explanation of the various tasks). These tables are quite helpful for someone like me who's unfamiliar with the field.
* The test set design is systematic and well designed.
* A good group of existing models are examined. Informative results are provided.

Weakness:
* I don't know if it's standard practice, but as far as I can tell, the Egyptian characters are not part of the dataset? Not sure if that's an oversell of the paper.
* the claim that "Models show a significant gap toward human-level generalization" is not experimentally evaluated. This sentence is briefly discussed in the caption of Figure 5, but i find the reasoning to be quite weak. The given dataset are based on simple arithmetic, so human are definitely having a very strong prior. By not evaluating the methods properly and isolate the effect of human prior, the claim seems unwarranted. For example, if everything is represented in Egyptian letter (perception level) and/or some new syntax / operations are involved, how can human extrapolate?

**Summary Of The Paper:**

The authors created a large dataset that examine seq2seq models' ability to perform perception, syntax and semantics learning. The test sets are carefully constructed to study interpolation and extrapolation in various combinations. A large set of existing models are examined to check their performances.

**Summary Of The Review:**

I like this paper and think this is above acceptance threshold. However, I'm waiting to see other reviewers's feedback (who probably are more familiar with the field than I am), before giving it a very strong endorsement.

---

> ### Author Response · Authors · 2022-11-19
> **Thank you for the constructive feedback!**
>
> We very much appreciate your efforts in reviewing our paper and providing insightful feedback. We are very encouraged to see your positive comments that "the dataset is quite interesting," "the presentation is quite clear," "the test set design is systematic," and "informative results are provided."
>
> We address your concerns as follows:
>
> > Egyptian characters are not part of the dataset?
>
> The Egyptian examples in Table 1 are used to illustrate how challenging this task is; they are NOT part of the dataset. We use Egyptian characters to avoid human prior, so that readers can understand the same level of challenges a computer program would encounter to solve this task.
>
> Of course, making an additional one with Egyptian characters is not difficult; one simply needs to replace all the symbols in HINT with scanned symbols of Egyptian characters. We will consider this option in the future.
>
> > The claim that "Models show a significant gap toward human-level generalization" is not experimentally evaluated.
>
> Thank you for pointing it out! To experimentally support the claim that "models show a significant gap toward human-level generalization when tested with **new concepts in a few-shot setting**," we conduct a pilot human study to evaluate the human performance in the few-shot learning experiment. Results are as follows: (the performance of LSTM and Transformer are extracted from Table 6 for comparison):
>
> | Concept | LSTM | Transformer | Human |
> |---------|------|-------------|-------|
> | x       | 42.8 | 52.8        | 95.0  |
> | y       | 35.4 | 50.7        | 100.0 |
> | a       | 33.8 | 46.4        | 97.5  |
> | b       | 31.1 | 42.6        | 77.5  |
> | c       | 21.9 | 27.3        | 90.0  |
> | d       | 24.7 | 30.4        | 60.0  |
> | Overall | 31.6 | 41.7        | 86.7  |
>
> The above results suggest a significant gap between the best model performance and human performance across all six concepts. We will add the above study in the revised draft.
>
> The human study setting:
> 1. Ten subjects are tested.
> 2. These six concepts are unknown to subjects to reduce the human prior as much as possible.
> 3. For each concept, the human subjects are asked to figure out its meaning from 10 training examples and answer 4 test questions.
>
> In the future, we plan to improve this pilot study (more subjects and more metrics to monitor human performance) and seek more insights into how well humans can learn new concepts in a few-shot manner.
>
> We hope the above explanations can answer your questions and clarify your concerns. Please let us know if there is any further question!

---

### Official Review · Reviewer_31D3 · 2022-10-24

**Confidence:** 4
**Correctness:** 4
**Technical Novelty And Significance:** 4
**Empirical Novelty And Significance:** 4
**Recommendation:** 8

**Clarity, Quality, Novelty And Reproducibility:**

**Clarity**
The paper is generally well written with well illustrated figures and informative tables. Table 2 is particularly helpful. The bullet points in Section 5 are also quite useful to guide the reader.

**Quality**
The decisions behind the construction of HINT are well justified. Experiments are comprehensive, with good choices of baselines and comprehensive additional details in the appendix.

**Originality**
The proposed dataset appears quite novel to my knowledge. Although on the surface it may appear similar to some previous datasets, the authors convincingly argue in Section 2 that HINT is designed to be significantly more challenging than prior datasets. As noted above, a particularly compelling aspect of the proposed dataset is its relative simplicity.



**Strength And Weaknesses:**

**Strengths**
The proposed dataset is simple to construct, yet very challenging which is a major plus point. As illustrated in Table 2, the dataset is categorically different from prior work since it tests systematic generalization over perception, syntax and semantics. The dataset has different levels of difficulty corresponding to different levels of generalization. Finally, a major strength of the dataset is the ability to test the acquisition of new concepts.

The empirical results are also well conducted. Strong baselines are tested on the task, including GPT-3, which provides confidence that the proposed benchmark is actually difficult to solve. The scaling experiments are also quite valuable since they demonstrate that the task can't be solved simply by using more data and computation. In my view, the few-shot learning results in section 4.2 are the most compelling experiments since they illustrate the difficulty of arguably the most challenging aspect of the task (namely, generalizing over semantics).


**Weaknesses**
Overall, the weaknesses of the paper are relatively minor. One concern is regarding the separation between syntax and semantics. In the proposed task, this separation is clear, but in other tasks it seems less clear. For instance, to what extent do solving tasks in CLEVR require understanding the semantics of objects in the scene? The authors appear to argue that semantics are not required, but justifying why this is the case would be helpful.

Also, the authors effectively test broad generalization over syntax and semantics. However, the way they test generalization over perception seems to be simply using different handwritten examples of the same symbols in the test set compared to the training set. Would it be possible to test a broader level of generalization over perception as well? For example, the authors could replace the symbols with the hieroglyphics from table 1 and test the learning system's ability to quickly learn and generalize to the new symbols.

It would also be helpful to understand how the models generalize separately on syntax and semantics. For instance, the authors may want to test generalization over syntax while keeping semantics fixed by testing on the same sequences as observed during testing but using different numbers (0, 1,... 9). This would be easier than the SS set proposed by the authors, but would still shed light on how the models operate.

**Summary Of The Paper:**

The paper proposes a new language-like dataset, HINT, that requires a learning system to map handwritten arithmetic expressions to numbers. In contrast to prior datasets, HINT require learning systems to learn perception, syntax and semantics. Experiments reveal that current LSTM and Transformer based architectures are unable to generalize well on the problem, and scaling experiments indicate that an infeasible amount of data and parameters are needed to perform well on the task.

**Summary Of The Review:**

The proposed dataset appears significantly more challenging than prior datasets, which is backed up by extensive experiments by the authors. The scaling experiments suggest that generalizing on this task will require new algorithmic or architectural innovations rather than merely larger scale. If this dataset encourages the development of new methods that can simultaneously learn perception, syntax and semantics, that could produce a very large impact in the field. Thus, I recommend acceptance.

---

> ### Author Response · Authors · 2022-11-19
> **Thank you for the constructive feedback!**
>
> We very much appreciate your efforts in reviewing our paper and providing insightful feedback. We are very encouraged to see your positive comments that "the proposed dataset appears quite novel," "the paper is generally well written," "the decisions behind the construction of HINT are well justified," and "experiments are comprehensive."
>
> We address your concerns as follows:
>
> > The separation between syntax and semantics seems less clear. For instance, to what extent do solving tasks in CLEVR require understanding the semantics of objects in the scene?
>
> Thank you for pointing it out! In Table 2, "semantics" represents if atomic concepts afford functional semantics in reasoning. In CLEVR, answering the questions requires parsing the questions into programs (syntax) and grounding the images on these programs (perception), while the objects themself do not afford any functional semantics. The task in CLEVR only involves objects' visual appearance, including shape, size, material, and spatial location, which are perceptual tasks. Hence, semantics is essentially not required in CLEVR.
>
> > A broader level of generalization over perception, e.g., replace the symbols with the hieroglyphics and test the ability to quickly learn and generalize to the new symbols.
>
> Good suggestion! This suggestion belongs to **domain generalization** (hieroglyphics is a different visual domain from handwritings), which is undoubtedly more challenging than our current setting for perception generalization (same domain, unseen instances). Studying perceptual generalization across domains is an important future direction for our work.
>
> > How the models generalize separately on syntax and semantics
>
> Per the suggestion, we evaluate the `Transformer rel.uni.` model from Table 5 on two new test subsets to understand how the models generalize separately on syntax and semantics:
> 1. semantics fixed: all the intermediate values are seen during training, but the tree structures are different. The Transformer rel.uni. model obtains an accuracy of 97.7%.
> 2. syntax fixed: all the tree structures are seen during training, but the numbers are different. The Transformer rel.uni. model obtains an accuracy of 97.3%.
>
> As expected, the `Transformer rel.uni.` model obtains high accuracies on both subsets and has slightly better generalization (interpolation) over syntax than semantics.
>
> We hope the above explanations can answer your questions and clarify your concerns. Please let us know if there is any further question!

---

### Official Review · Reviewer_52ig · 2022-10-25

**Confidence:** 3
**Correctness:** 3
**Technical Novelty And Significance:** 3
**Empirical Novelty And Significance:** 3
**Recommendation:** 6

**Clarity, Quality, Novelty And Reproducibility:**

The paper is very well-written. The results are clearly presented. However, important related works / baselines are missing to further establish its novelty.

**Strength And Weaknesses:**

HINT is a well-designed dataset with simple data yet complex generation tasks. I like the minimalistic design to keep the generalization study clean across 3 different domains, this way we can diagnose a model in more depth. The paper includes evaluations of several state-of-the-art neural models, including variations of recurrent nets (LSTM, GRU), transformers, and GPT-3. Empirical study shows these neural models' capabilities of generalization across perception, syntactics and semantics, revealing a general lack of power on the latter two domains.This paper also provides insightful results on few-shot generalization of new arithmetic concepts.

To me the weakness of this paper is also obvious. The baselines are too simple and only concentrate on neural networks, while on the other hand semantic parsing has been long-studied by the NLP community ([1] as an example). Even within the domain of neural networks, there are models designed for generalization already been published for a long time (such as [2]). It is very unfortunate that these related works are missing from the discussion.

[1] Guu et al. 2017 "From Language to Programs: Bridging Reinforcement Learning and Maximum Marginal Likelihood"
[2] Andreas et al. 2017 "Neural Module Networks"

**Summary Of The Paper:**

This paper introduces a new dataset: HINT for studying machine reasoning across the joint domain of perception, syntax and semantics. HINT is a light-weighted dataset based on hand-written style digit arithmetic. Despite its simplicity, this paper introduces various generalization tasks over these three domains and demonstrates that state-of-the-art neural models perform poorly on these tasks. HINT sheds light on future studies of full-spectrum generalizability for machine reasoning.

**Summary Of The Review:**

My current assessment to this paper is borderline accept, conditioned on the authors constructive response regarding related fields / baselines. I won't object if this paper is rejected because of weakness on these aspects.

---

> ### Author Response · Authors · 2022-11-19
> **Thank you for the constructive feedback!**
>
> We very much appreciate your efforts in reviewing our paper and providing insightful feedback. We are very encouraged to see your positive comments that "HINT is a well-designed dataset," "insightful results on few-shot generalization," and "the paper is very well-written."
>
> We address your concerns as follows:
>
> > Missing baselines and related works from semantic parsing and neural module networks.
>
> Thank you for pointing out! The two papers mentioned here belong to neural-symbolic methods. Both of them map the input (either a natural language instruction [1] or a question [2]) to an executable program and execute the program by an execution engine, which is either provided beforehand [1] or learned via neural modules [2]. The domain-specific language (DSL) for the program space is required. Of note, we have already discussed this line of work---neural-symbolic methods on systematic generalization---in **the last paragraph of the "Related Work" section**. These neural-symbolic methods require non-trivial domain-specific knowledge; we leave the exploration of these sophisticated methods for future work.
>
> Still, we acknowledge the significance of establishing neural-symbolic baselines on HINT. Therefore, as preliminary results, we evaluate the RANDOMER model [1] and the neural module network (NMN) [3] by adapting their public codebases to HINT.
>
> In HINT, the domain-specific language (DSL) for programs is defined as Reverse Polish Notation, a.k.a. postfix notation. We implement the program generator using LSTM with three encoder layers and one decoder layer. The program generator takes as input a handwritten image and generates a postfix expression. Of note, RANDOMER requires an execution engine to calculate the results of postfix expressions, which means that the semantics of concepts are given and not learned. Similarly, although NMN learns an execution engine via neural modules, it still requires extra annotations for programs.
>
> The results are as follows:
> |    Model    |   I  |  SS  |  LS  |  SL  |  LL  | Avg. |
> |-------------|:----:|:----:|:----:|:----:|:----:|:----:|
> | Transformer | 88.4 | 86.0 | 62.5 | 10.9 | 19.1 | 53.1 |
> |   RANDOMER  | 85.1 | 73.2 | 40.2 | 51.6 | 38.5 | 58.6 |
> |     NMN     | 86.3 | 85.7 | 66.3 | 11.3 | 20.4 | 55.4 |
>
> The above results demonstrate that HINT is very challenging for current neural-symbolic models, echoing one of the motivations. Of note, although RANDOMER obtains a high accuracy on SL, it is mostly because the ground-truth semantics are encoded in the execution engine beforehand. We want to emphasize that it is unfair to directly compare RANDOMER and NMN with neural baselines like Transformer, because RANDOMER and NMN require domain-specific knowledge for the model design and extra annotations for the model training.
>
> We will include the above results in the revised draft and establish more neural-symbolic baselines in the future.
>
> [1] Guu et al. 2017 "From Language to Programs: Bridging Reinforcement Learning and Maximum Marginal Likelihood."
>
> [2] Andreas et al. 2017 "Neural Module Networks."
>
> [3] Johnson et al. 2017 "Inferring and Executing Programs for Visual Reasoning"
>
> We hope the above response can resolve your concerns. Let us know if there is any further question or suggestion!

---

### Author Response · Authors · 2022-11-19
**General comments for all reviewers and the summary of revision**

We are grateful for your constructive comments and helpful feedback. We are very encouraged by the acknowledgment that
1. Our paper is well written (all reviewers);
2. The proposed HINT dataset is well-designed, simple to construct, yet very  challenging (Reviewers 52ig, 31D3, LVqC);
3. The empirical results are well-conducted and informative (Reviewers 31D3, LVqC, q55G).

Following your suggestions, we have improved our work and revised the paper accordingly. The revised texts are highlighted in blue. Please refer to the newest draft for details. Below, we summarize the revision highlights:
1. We have included neural-symbolic baselines from semantic parsing (Guu et al. 2017) and neural module networks (Johnson et al. 2017). Details and results are in Appendix, Section A.4. (Suggested by Reviewer 52ig)
2. We have added a pilot human study to evaluate human performance in the few-shot learning experiment. The results are appended to Table 6, whereas the setup of the human study is described in Appendix, Section A.5. (Suggested by Reviewer LVqC)
3. Revisions regarding writing. (Suggested by Reviewers 31D3 & q55G)

We appreciate all the suggestions made by reviewers to improve our work. We look forward to answering your follow-up questions.

---

### Decision · Program_Chairs · 2023-01-20

**Decision:**

Accept: notable-top-25%

**Justification For Why Not Higher Score:**

It provides a challenging and needed task, but not with a ground-breaking methodology.

**Justification For Why Not Lower Score:**

Given the overall positive reviews from all reviewers, I would like to stress the importance of testing models that are expected to learn all 3 levels end-to-end, without intermediate supervision, in the era of LLMs. I think the work should be at least a spotlight.

**Metareview: Summary, Strengths And Weaknesses:**

This paper introduces a new dataset: HINT for studying machine reasoning across the joint domain of perception, syntax and semantics. In contrast to prior datasets, HINT require learning systems to learn perception, syntax and semantics. Experiments reveal that current LSTM and Transformer based architectures are unable to generalize well on the problem, and scaling experiments indicate that an infeasible amount of data and parameters are needed to perform well on the task.

+ The proposed dataset is simple to construct, yet very challenging which is a major plus point.
+ Strong baselines are tested on the task, including GPT-3, which provides confidence that the proposed benchmark is actually difficult to solve.

Overall, the weaknesses of the paper are relatively minor.

Given the overall positive reviews from all reviewers, I would like to stress the importance of testing models that are expected to learn all 3 levels end-to-end, without intermediate supervision, in the era of LLMs. I think the work should be at least a spotlight.

**Note From Pc:**

if the above contains the word "oral" or "spotlight" please see: "oral" presentation means -> notable-top-5% and "spotlight" means -> notable-top-25%. As stated in our emails, we are disassociating presentation type from AC recommendations

**Summary Of Ac-Reviewer Meeting:**

N/A